# Research on a virtual-real fusion experimental system for the cutting part of a boom-type roadheader

**Jianzhuo Zhang[1], Ce Chen[1]\*, Xiaoyu Ding[2], Tao Wang[1], Chuanxu Wan[1], Wenliang Li[3]**

1 School of Mechanical Engineering, Liaoning Technical University, Fuxin City, Liaoning, China, 2 Zoucheng Yicheng machinery Co., LTD, Zoucheng City, Shandon, China, 3 Shandong Yankuang Intelligent Manufacturing Co., Zoucheng City, Shandong, China

\* 994745165@qq.com

## Abstract

Aiming at the problems of high safety risks, economic costs, and inefficiency in experimental research on boom-type roadheaders, this study proposes a virtual-real fusion experimental system for the cutting module. This system incorporates functions including digital modeling of coal-rock, simulation of mechanical properties of the cutting unit, and integration of virtual and physical experiments. To address the challenge of obtaining cutting tooth loads at coal-rock interfaces, a discretized digital coal-rock volume construction method is proposed. For rapid mechanical performance simulation of the cutting unit, a chain-type digital mapping body construction method is developed. Through deep learning, numerical simulation, and digital twin technologies, a virtual-real fusion platform was established, enabling virtual experiments to dominate the calibration of physical experiments. The system is capable of simulating pose variation and vibration trends of the entire machine during cutting. The minimum average error for stress at the cylinder base is 4.44 MPa, with a virtual-real system connection period under 100 ms. Based on this system, a reinforcement learning training environment was developed. Using Deep Deterministic Policy Gradient (DDPG), the control of the cutting unit was optimized to achieve low-stress state cutting, verifying the system's feasibility.

## 1. Introduction

In the field of deep underground engineering and tunneling, the boom-type roadheader is a highly flexible and efficient excavation equipment, which is widely used in underground space excavation projects such as tunnels and mine roadways [1]. Its construction process features high levels of automation, labor-saving, fast construction speed, and good control over excavation [2]. At present, research on the measurement and control of the operating posture of the boom-type roadheader, the design and development of the intelligent cutting system, and the analysis and

**Data availability statement:** All relevant data are within the manuscript and its Supporting Information files.

**Funding:** All the funds for this project are provided by National Key Research and Development Program of China - Intelligent impact support technology and equipment foiimpact hazardous tunnels and engineering demonstration(2022YFC3004605). The funder is Zhang Jianzhuo. There was no additional external funding received for this study.

**Competing interests:** The authors declare that they have no known competing financial interests or personal relationships that could have appeared to influence the work reported in this paper.

optimization of mechanical properties are important guarantees for improving the safety and efficiency of tunneling operations [3,4]. For instance, intelligent control of rotational and swing speeds during excavation maintains consistent cutting torque and structural stress, thereby minimizing equipment vibration and wear under variable coal-rock conditions [5]. The integration of image processing and deep learning further enhances system intelligence while reducing operational safety risks [6]. Traditional research methods relying on full-scale physical prototypes face limitations including prolonged development cycles, high costs, and challenges in replicating extreme operating conditions. Studies involving pose recognition and adaptive control which depend on sensor data and simulations confront additional hurdles: adverse environmental factors at tunnel faces, difficulties in acquiring real-time coal-rock properties and equipment status data, and stringent safety protocols collectively escalate experimental durations, costs, and risks [7–9]. Overcoming these cost and efficiency barriers remains a pivotal challenge in advancing intelligent roadheader development.

Current methodologies primarily involve physical experiments and numerical simulations. While physical experiments provide reliable data, their implementation involves repetitive tasks, accelerated equipment degradation, and high temporal-economic costs, limiting multi-parameter combinatorial validation. For instance, fabricating coal-rock specimens and deploying sensor networks demand substantial labor and material investments [10]. Data-intensive approaches like deep learning face scalability challenges due to the prohibitive costs of acquiring labeled datasets through full-scale underground or surface cutting experiments [11–13]. Although numerical simulations reduce costs through virtual data generation, two critical limitations persist: (1) accuracy degradation from idealized boundary conditions, and (2) inadequate representation of system-level behaviors when simulating isolated components [14]. System-scale simulations further require complex coupling across multiple software platforms, compounded by the lack of real-time cross-validation between numerical and physical domains [15–18]. This decoupling prevents dynamic error correction, necessitating post-hoc experimental verification of simulation datasets.

Current experimental approaches predominantly employ bench-scale or reduced-dimension prototypes to mitigate the complexity of boom-type roadheader testing. Existing platforms simulate operational states through cutting head pose recognition, control system emulation, and concrete cutting modules. However, material-based simulations (e.g., concrete/gypsum-coal mixtures) lack real-time tunability of mechanical properties, while requiring extensive manual calibration to approximate coal-rock behaviors [19]. Although theoretical modeling and virtual simulations partially alleviate these issues, physical validation remains mandatory [20]. Integrating physical prototypes with theoretical models and virtual simulations into a unified virtual-real testing system could systematically mitigate experimental complexity. Specifically, virtual sensors enable automated data acquisition during testing cycles, circumventing the need for intrusive physical instrumentation. This integration establishes a foundational platform for advanced roadheader research with enhanced self-sensing capabilities.

Digital twin technology represents a prominent research field in virtual-real system integration. This technology enables real-time synchronization between physical entities and their virtual counterparts through bidirectional data exchange [21–23]. Despite these advances, current implementations predominantly focus on pose monitoring and control of boom-type roadheaders, while neglecting integrated mechanical simulation and experimental validation [24–26]. Deep learning exhibits exceptional nonlinear regression capabilities, allowing it to approximate nonlinear interactions between boundary constraints and structural response fields when trained on numerical datasets [27,28]. Such models facilitate the generation of reduced-order simulations and digital twin mappings, thereby accelerating mechanical analysis [29–32]. By coupling physical prototypes with their digital twins, a hybrid experimental framework emerges, unifying virtual and physical testing modalities under a single architecture.

This study aims to develop a cutting module experimental system for boom-type roadheaders, establishing a virtual-dominant physical calibration paradigm to reduce experimental costs, enhance dataset accuracy, and improve system intelligence. The proposed system integrates five core functionalities:

1) Constructing digital coal-rock models (homogeneous/heterogeneous).

2) Identifying and fitting pick-coal-rock interface coupling information.

3) Real-time monitoring of physical entities and their digital twins.

4) Dynamic simulation of cutting kinematics with whole-machine multi-modal perception.

5) Cross-domain fusion of virtual-real datasets for accuracy enhancement.

The system implementation integrates four core technologies: numerical simulation, sensor networks, deep learning, and visual rendering [33–36]. This study establishes a virtual-real fusion framework where virtual experimentation dominates the testing process, supplemented by targeted physical validations. The Deep Deterministic Policy Gradient (DDPG) algorithm serves as a validation benchmark to verify system efficacy.

The main contributions are:

1) This paper presents a virtual-real experiment system of boom-type roadheader, proposing a experimental framework prioritizing physical validation supplemented by extensive virtual experiments.

2) A deep learning-based method constructs discrete coal-rock digital models, resolving the digitization of cutting tooth-coal-rock coupling relationships during virtual cutting. This enables simulation of the roadheader's mechanical properties under diverse coal-rock conditions.

3) To digitize structural and mechanical characteristics of physical roadheader equipment, Reduced Order Model (ROM) technology builds component-level digital mappings. Chained transfer of coupling information reduces dataset construction complexity, while D-H coordinate transformations enhance full-machine digital mapping accuracy. Real-time output of position, posture, and mechanical properties is achieved.

4) Application of the Deep Deterministic Policy Gradient (DDPG) algorithm in the virtual-real fusion system reduces cutting-part stress values, verifying system feasibility.

(Fig 1) details the six-step workflow:

1) Construction of the physical testbed for the cutting assembly.

2) Segmentation of the cutting module into structural subcomponents and definition of boundary interaction parameters.

3) Dataset generation for input-output mappings via numerical simulation and experimental analysis.

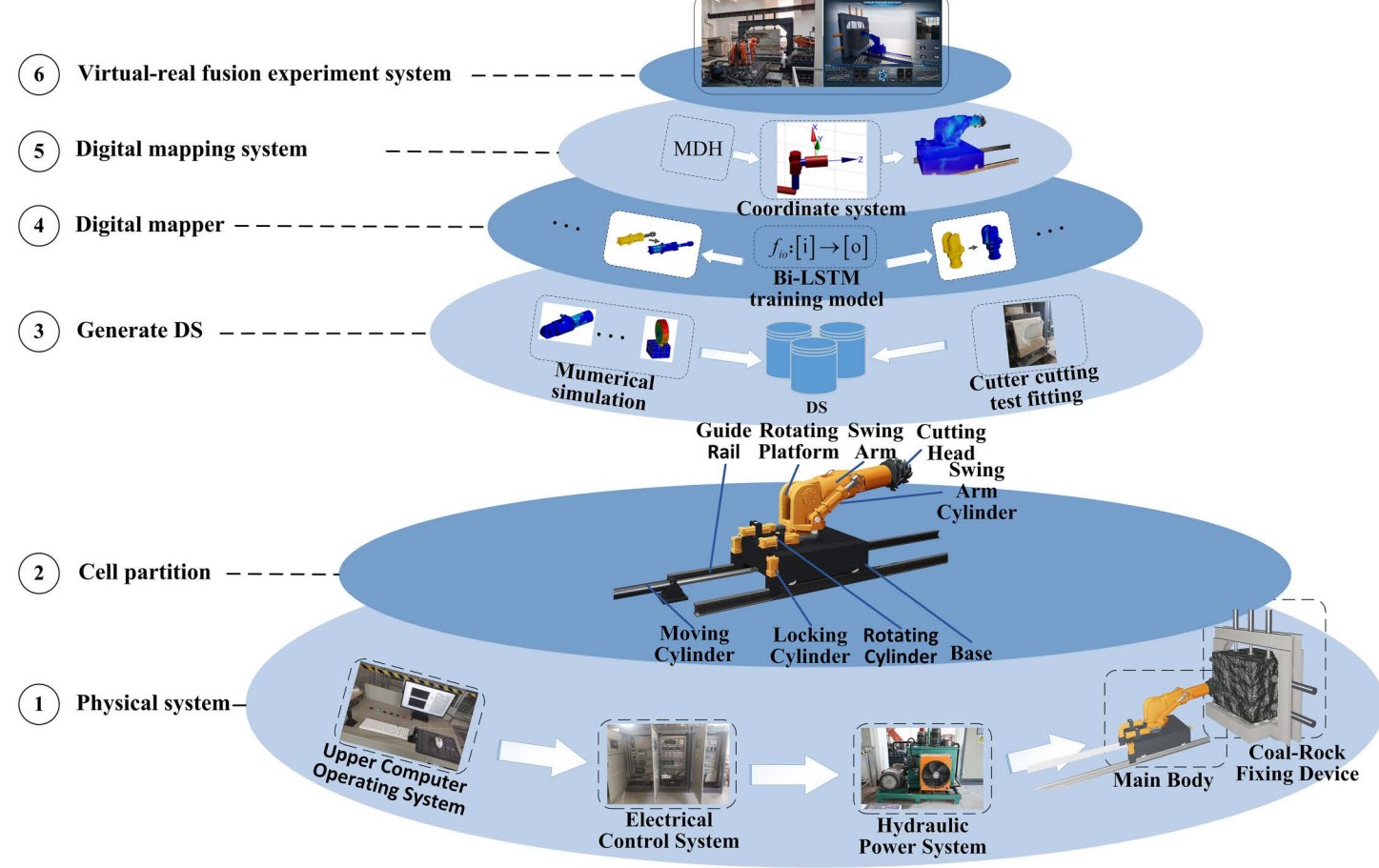

**Fig 1. System construction process.**

https://doi.org/10.1371/journal.pone.0330547.g001

4) Bi-LSTM networks train and fit each dataset to establish digital twins for structural components.

5) Coordinate transformation matrices for each structural part are computed and assembled to construct the full-machine digital twin system.

6) Industrial communication protocols connect the digital twin system with the cutting module's electrical control system, completing the virtual-real fusion testing platform.

The remainder of this paper is structured as follows: Section 2 Proposes a discrete coal-rock cubic unit model and its construction methodology. Section 3 Presents a chain-based digital twin framework for full-machine simulation with precision enhancement. Section 4 Details the virtual-real fusion system and validates accuracy through cutting experiments. Section 5 Verifies system feasibility using the Deep Deterministic Policy Gradient (DDPG) algorithm. Section 6 Concludes with research findings and future work.

## 2. Identification method of load on cutting tooth of coal-rock

The cutting drum drives the cutting teeth to fracture coal-rock masses, constituting the core loading mechanism of the boom-type roadheader's cutting assembly. Consequently, coal-rock loads on cutting teeth serve as critical input

parameters for the virtual experimental system. The digital twin construction for coal-rock loading hinges on establishing input-output datasets, defined as:

$$f_{io} : [i] \rightarrow [o] \tag{1}$$

Here, $f_{io}$ is the mapping relationship of coupling information from coal to cutting teeth. $[i]$ is the input dataset set. $[o]$ is the output dataset set. Deep learning enables nonlinear fitting of input-output dataset mapping relationships, generating a digital twin for coal-rock loads on cutting teeth. However, affected by the rotation of the cutting head and the crushing of coal-rock, the information of coal-rock or the load of the cutting teeth are difficult to be directly digitized. The focus of this study is to determine the input and output data parameters, establish cutting tooth load datasets, and construct a digital coal-rock model. The following sections detail: (1) discretization methods for coal-rock cubic units, and (2) dataset construction approaches through numerical simulation and experimental studies.

## 2.1. Coal-rock discrete cubic unit

The experimental system focuses on the cutting assembly of the boom-type roadheader. Given the heterogeneous and friable nature of coal-rock masses, fully replicating their physical characteristics is infeasible. Therefore, only a digital representation of load outputs from coal-rock cutting teeth is established. To enable virtual simulation of homogeneous or heterogeneous coal-rock, the coal-rock mass is discretized into cubic coal-rock unit $m_5$. Each cubic coal-rock unit $m_5$ is associated with a mapping relationship $f_{io}:[i] \rightarrow [o]$. By using deep learning to fit the cube coal-rock unit $m_5$ with the load information $f_{io}$ of the cutting teeth, a digital mapping body for the load of the cube coal-rock on the cutting teeth is constructed. Finally, multiple identical or different digital mapping bodies are used to build the overall digital mapping body for the load of the coal-rock on the cutting teeth.

During cutting tooth penetration into coal-rock, beyond the mechanical properties of the coal-rock materials, the mechanical state of the cutting tooth depends primarily on contact area and contact pressure. Specifically: Contact area is governed by the penetration depth and attack angle of the cutting tooth; Contact pressure depends on the cutting speed and damage coefficient of the coal-rock unit. Therefore, in this study, parameters such as the penetration depth, attack angle, cutting speed, damage coefficient are taken as parameters in $[i]$. As shown in (Fig 2). The penetration depth and attack angle are derived from the homogeneous transformation matrix of the cutting tooth relative to the coal-rock element's local coordinate system:

$$T = \begin{bmatrix} n & o & a & P \end{bmatrix} = \begin{bmatrix} n_x & o_x & a_x & X_0 \\ n_y & o_y & a_y & Y_0 \\ n_z & o_z & a_z & Z_0 \\ 0 & 0 & 0 & 1 \end{bmatrix} \tag{2}$$

Here, $n$, $o$, $a$ are the element vector of the coordinate axis of the cutting tooth coordinate system, $P$ is the position array of the cutting tooth.

Cutting tooth speed is calculated by 4 degrees of freedom velocity Jacobi:

$$\dot{X} = J(q)\,\dot{q} \tag{3}$$

Here, $\dot{X}$ is the generalized velocity of the cutting tooth, $J(q)$ is the Jacobian matrix, $\dot{q}$ is joint velocity.

The damage coefficient of the cubic unit is determined by the number of cutting cycles and the damaged volume. The cutting cycles are defined as complete traversals of a single cutting tooth through the unit. Due to the randomness of the damaged areas in the coal-rock, and the influence of cutting depth, angle, speed, and the damage coefficient of the

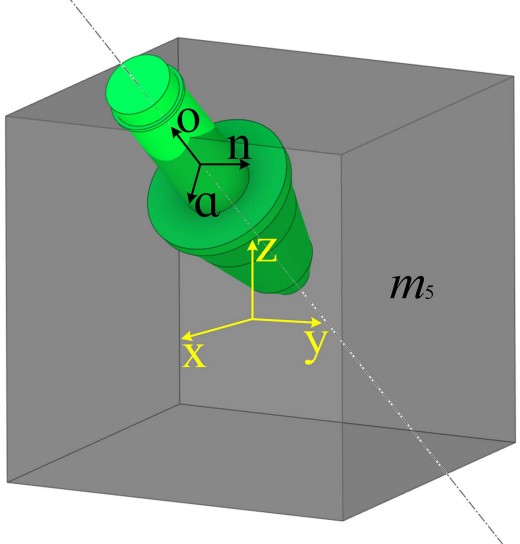

**Fig 2. Cutting tooth cutting position diagram.**

coal-rock unit at the boundary of the cutting coal-rock cube unit, it needs to be obtained through deep learning fitting of numerical simulation and cutting test data. Therefore, the damage coefficient of the cut cube coal-rock unit $m_5$ in $f_{io}$ is both an input parameter and an output data. During the process of constructing the training dataset, the damage coefficient of the coal-rock before being cut was recorded as the input data, while the damage coefficient of the coal-rock after being cut was used as the output data. During the process of constructing the training dataset, the damage coefficient of the coal and rock before being cut was recorded as the input data, while the damage coefficient of the coal and rock after being cut was used as the output data. During the simulation of the cutting process, at the initial moment when no cutting occurred, $m_5$ was 1. The damage coefficient of the cubic coal-rock units that were cut is:

$$m_5 = 1 - \eta_1 - \eta_2 - \cdots - \eta_n \tag{4}$$

Here, $\eta_n$ is the proportion of damaged area in the $n$ cut.

The cutting tooth of the cubic unit of coal-rock should also consider whether there are unfinished cubic units of coal-rock in the boundary space. Therefore, it is necessary to add the boundary space information matrix $K_B$ to the input information set of the cubic coal-rock unit to the digital mapping of the pick-load. Since hard coal-rock is not prone to large-scale fragmentation, the coal-rock that is not adjacent to the cubic coal-rock unit $m_5$ has a relatively small impact on the cutting process of the teeth on unit $m_5$. Therefore, the degree of damage of the coal-rock that is not adjacent to unit $m_5$ was not introduced into the calculation. The boundary spatial information matrix of coal-rock unit takes the coal-rock unit $m_5$ as the center, with 26 surrounding cubic units as the boundary information elements. as shown in (Fig 3).

$$K_B = \left[ \begin{array}{ccccccccc} l_1 & l_2 & l_3 & m_1 & m_2 & m_3 & r_1 & r_2 & r_3 \\ l_4 & l_5 & l_6 & m_4 & m_5 & m_6 & r_4 & r_5 & r_6 \\ l_7 & l_8 & l_9 & m_7 & m_8 & m_9 & r_7 & r_8 & r_9 \end{array} \right] \tag{5}$$

Here, $l_{1-9}$, $m_{1-9}$, $r_{1-9}$ are the damage coefficient of coal-rock element.

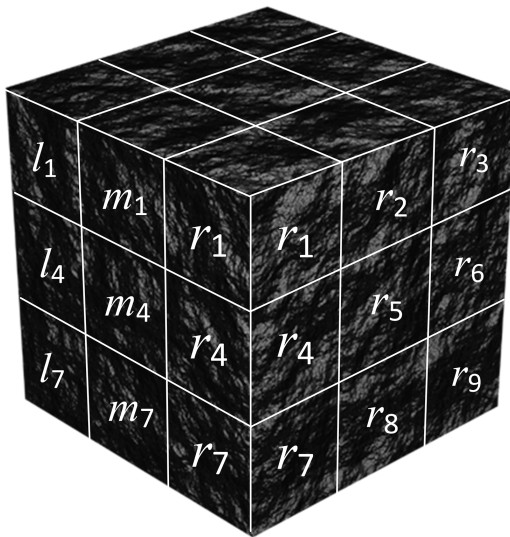

**Fig 3. Partition of information elements around coal-rock unit $m_5$.**

The output data of the load of the coal-rock unit on the cutting tooth include the stress and deformation data of the cutting tooth $\sigma$, The joint force of the cutting tooth on the cutting tooth seat $F_{jm}$ and the proportion of the damaged area $\eta_n$ of the coal-rock unit for the $n$ cutting. In summary, the input and output data set of the mapping relationship between the load of cutting tooth of cubic coal-rock unit is as follows:

$$\begin{cases} [i] = \begin{bmatrix} K_B & T & \dot{X} \end{bmatrix} \\ [o] = \begin{bmatrix} \sigma & F_{jm} & \eta_n \end{bmatrix} \end{cases} \tag{6}$$

Here, $T$ is the position and homogeneous transformation matrix of cutting tooth, $\sigma$ is the stress and deformation data of cutting tooth, $F_{jm}$ is the joint force of the cutting tooth to the tooth seat.

## 2.2. Input/output data set construction

The input and output data sets are constructed using two methods: numerical simulation and cutting test. Both of them use the method of single tooth cutting to perform rotating cutting of cubic unit coal-rock at various cutting angles. The cut cubic coal-rock unit is divided into 27 small cubic units, and each small cubic unit is regarded as once cut coal-rock unit $m_5$, and the damage coefficient of the no coal-rock area around $m_5$ element is 0. In the cutting process, the cutting tooth and each small cubic unit are constructed with input and output data sets, that is, each cubic coal-rock can generate 27 sets of input and output data sets, as shown in (Fig 4).

The process of cutting coal-rock with cutting tooth is simulated numerically by using the element failure method in display dynamic analysis. The control parameters include speed and cutting angle. The extracted data includes: the cutting tooth axis coordinates and rotation speed within a fixed time interval, which are used to calculate the cutting tooth homogeneous transformation matrix $T$ and cutting tooth speed $\dot{X}$; The ratio of failure grid to the number of coal-rock grid is used to calculate the proportion of damaged areas after each cut; The stress and deformation matrix $\sigma$ is generated from the cutting tooth stress deformation data; The vector force matrix $F_{jm}$ is generated by The joint force of the cutting tooth on

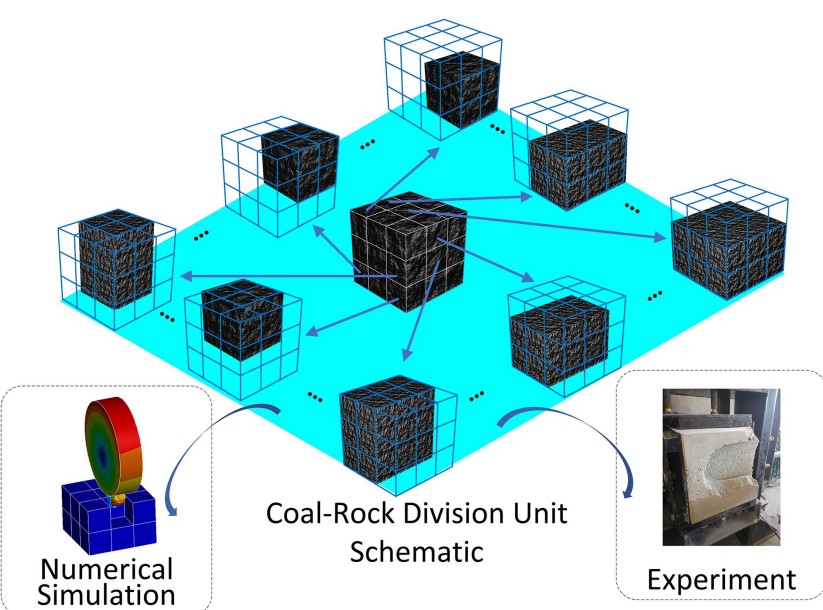

**Fig 4. Input/output data set construction.**

the cutting tooth seat. In the cutting test, a single cutting tooth is used to cut cubic coal-rock unit, and the data extracted is the same as the numerical model.

After constructing the input and output datasets, data normalization is first applied. The datasets are divided into the training set, the validation set and the test set in a ratio of 7:2:1. Subsequently, the Deep Convolutional Neural Networks (DCNN) is employed for fitting training. The DCNN implementation uses TensorFlow with the following architecture: the number of network layers is set to 7 layers, the activation function adopts ReLU, and the loss functions all adopt mean square deviation functions. The trained model is deployed as a C++ executable module, forming the digital coal-rock volume. Finally, spatial stacking of multiple cubic unit digital twins constructs the complete digital coal-rock model.

## 3. The digital twin of the cutting part is built

After the load mapping relationship between coal-rock to cutting tooth is constructed, the same or different coal-rock cubic units can be used to build the homogeneous or heterogeneous coal-rock digital twin to form the coal-rock information matrix. The coal-rock information matrix contains the coordinates of each coal-rock element and the information of coal-rock material. In the virtual space, when the cutting drum overlapped with the coal-rock space, the output information of the coal-rock element to the load mapping body of the cutting part was output by each cutting tooth through the coordinate change and identification of the coal-rock material number, and was used as the load input information of the virtual experiment system of the cutting part of the boring machine. After the load data of the cutting part is obtained, it is necessary to build the load information into the digital mapping of the physical characteristics of the whole machine.

Reduced order model (ROM) technology is designed to reduce computational costs and speed up numerical simulation solutions. In this paper, the non-invasive reduced order model (NIROM) is used to reduce the order of the numerical simulation, and the digital mapping of the whole machine structure is generated to realize the rapid output of the structural mechanical property data. The main steps: first build the input-output data set through numerical simulation, then fit the input-output mapping relationship through deep learning, and finally deploy the training model. Among them, the input and output data set constructed by numerical simulation is the key influencing factor of the digital mapping of the whole

machine. Therefore, the construction method of the digital mapping of the physical characteristics of the whole machine will be discussed below to improve efficiency and ensure accuracy.

## 3.1. Chain digital twin

In order to improve the accuracy of the data set, the key structure part of the cutting part of the boom-type roadheader was first divided, the single structure part data set was constructed respectively, and the single structure part digital twin was trained to generate. Then, the digital twin of the cutting part was built by transferring mechanical parameters and linking each digital twin. During the numerical simulation of single structure part, the parts contact less, the solution time is short, and the data accuracy is higher than that of the whole machine. Since the physical state of each structure part changes rapidly during the cutting part experiment, the display dynamics method is used to numerically simulate each structure part, and the input parameter information, mesh node stress and deformation information and output parameter information of each structure part are extracted to establish a data set.

The cutting part is divided into cutting head, swing arm, rotating platform, base, guide rail, moving cylinder, locking cylinder, rotating cylinder and swing arm cylinder. The coupling of the digital mapping bodies of each structure part takes force as the transmission parameter. And forms a chain structure of the whole machine through multi-level transmission, as shown in (Fig 5). The swing arm cylinder and rotating cylinder are the main driving units, and their digital mapping bodies contain displacement and pressure input and output boundary parameters. The digital mapping of the cutting head is an external information input structure. Besides the speed and torque collected by the sensor as the boundary input parameters, the joint force of each cutting tooth should also be taken as the boundary input parameters. The moving cylinder does not participate in the cutting process. And is only responsible for the distance control between the cutting part and the coal-rock, so its digital twin only outputs the pose signal.

The attitude and mechanical states of each structural body comprise sensor-derived data and digital twin outputs. Among them, the sensor data are obtained from the displacement sensor and pressure sensor from the translational cylinder, the displacement sensor and pressure sensor from the rotating cylinder, the displacement sensor and pressure sensor from the swing arm cylinder, and the torque and speed sensor from the cutting head. Non-sensor data primarily consist the mechanical data and pose correction data of each structure, derived via component-level digital twins. To ensure output accuracy, input-output datasets must encompass diverse operating conditions. The input data set should cover all factors affecting outputs, as shown in (Fig 5). Numerical simulations under various working conditions are conducted by changing the input parameters to cover as many possible working conditions as possible. Mechanical states and pose corrections extracted from these simulations construct the datasets. Deep learning establishes the mapping between input parameters and output responses, enabling real-time prediction of the boom-type roadheader's mechanical states. This approach achieves real-time numerical simulation capabilities, providing stress and deformation data at grid nodes. The digital simulation employs the LS-DYNA solver, the solid adopts the Lagrangian, the oil adopts the Arbitrary Lagrangian Eulerian (ALE), and the joint adopts the Kinematic pair connection.

The stress and deformation of structural parts is the time sequence information, and the stress and deformation at the current moment is related to the stress and deformation in the past period of time. Therefore, the Bi-LSTM model combined with the forward Long Short Term Memory (LSTM) and backward LSTM is used to train the numerical simulation data, and the time sequence feature extraction training is carried out by integrating the stress and deformation data at the $t$ time and the before and after time. The mapping relationship between input parameter information and structural part grid node stress and deformation information is constructed. By two-way feature extraction, the information in the sample can be extracted more efficiently, and the features in the timing information can be comprehensively captured, so that the training model can converge faster and reach a lower loss value. The data set of stress and deformation information of each structural component generated by numerical simulation was normalized. The number of network layers was set to 4, the number of units was set to 11, Relu activation function was used, and the last layer was fully connected. The output

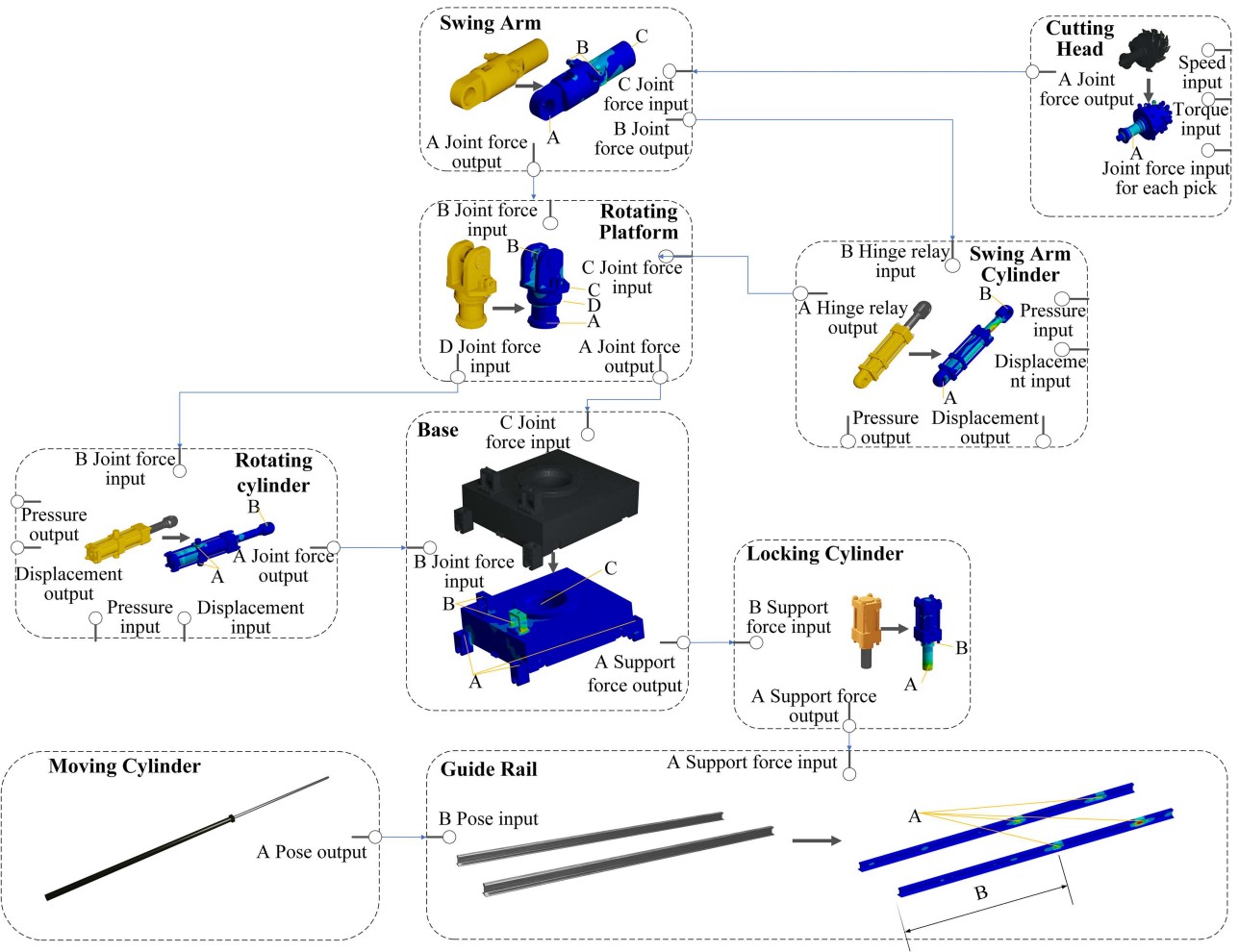

**Fig 5. Each key structural part digital twin.**

number was the number of grid nodes of each structural component, and the division ratio of training set and test set was 9:1. The Bi-LSTM models of all parts are run in Intel-11700k, NVIDIA-3080TI, and tensorflow-gpu2.4.0 environments. The model hyperparameters are default, and the mean absolute error function is used as the loss function. The epoch is set to 30000.

### 3.2. Construction and modification of kinematic equation

Single structure part construction method can reduce the difficulty of data set construction and improve the precision of digital mapping of the whole machine, but it can not control the spatial coordinates of each structure part, so it is necessary to establish the kinematic equation of the cutting part. The cutting part is an open kinematic chain mechanism, which includes four degrees of freedom: horizontal translation, horizontal rotation, vertical swing and cutting head rotation. The horizontal rotation angle and vertical swing angle are controlled by the rotating cylinder and the swing arm cylinder. The geometric relation between rotating cylinder displacement and horizontal swing angle and the geometric relation between swing arm cylinder displacement and vertical swing angle is constructed by the triangular function relation. D-H method

was used to construct the mapping relationship between the translational displacement, horizontal swing angle and vertical swing angle to the kinematic equation of the cutting part, and the digital mapping between the displacement of the active cylinder and the pose of the cutting part was formed.

The geometric relationship of the rotation and swing space of the cutting part is shown in (Fig 6). The rotation angle and swing angle can be obtained from the trigonometric function relationship:

$$\begin{cases} \theta_2 = \arccos\left(\frac{l_1^2+l_2^2-(x_2+l_3)^2}{2l_1l_2}\right) - \arccos\left(\frac{l_1^2+l_2^2-l_3^2}{2l_1l_2}\right) \\ \theta_3 = \arccos\left(\frac{l_4^2+l_5^2-(x_3+l_6)^2}{2l_4l_5}\right) - \theta_a \end{cases} \tag{7}$$

Here, $x_2$ is the displacement of the rotating cylinder, $x_3$ is the displacement of the swing arm cylinder, $\theta_2$ is the horizontal swing angle, $\theta_3$ is the vertical swing Angle, $\theta_a$ is the swing arm axis and $l_4$ angle, $l_1$ is the distance between the rotating center of the rotating cylinder and the rotating center of the rotating platform, $l_2$ is the rotating distance between the end of the piston rod of the rotating cylinder and the center of the rotating platform, $l_3$ is the initial distance between the rotating center of the rotating cylinder and the end head of the rotating cylinder piston rod, $l_4$ is the distance between the rotating center of the swing arm and the end of the piston rod of the swing arm cylinder, $l_5$ is the distance between the rotating center of the swing arm and the rotating center of the swing arm cylinder, $l_6$ is the initial distance between the rotating center of the swing arm cylinder and the end of the piston rod of the swing arm cylinder.

The kinematic equation of the boom-type roadheader's cutting module are derived using the Modifie-Denavite-Hartenberg (MDH) method, as shown in (Fig 7), with the fixed hinge seat of the moving cylinder serving as the global coordinate origin $O_0$ $(0, 0, 0)$. The MDH parameters and joint variables of each structure are listed in Table 1. The movement of the cutting part of the boom-type roadheader is controlled by the displacement $x_1$ of the moving cylinder, the displacement $x_2$ of the rotational cylinder, the displacement $x_3$ of the swing arm cylinder and the rotation angle $\theta_4$ of the cutting head. The pose state of the cutting part is calculated by combining the displacement change data output by the digital twin of each cylinder structure.

The kinematic equation of the base is obtained by calculating the displacement of the moving cylinder:

$$\begin{cases} [x_1] \rightarrow T_1 = A_1 = \begin{bmatrix} 1 & 0 & 0 & 0 \\ 0 & 0 & 1 & x_1 \\ 0 & -1 & 0 & 0 \\ 0 & 0 & 0 & 1 \end{bmatrix} \\ x_1 = x_1^c + \Delta x_1 \end{cases} \tag{8}$$

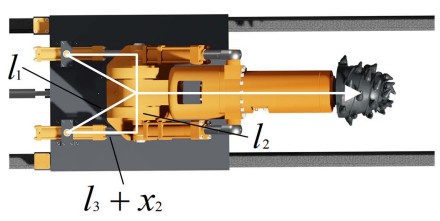

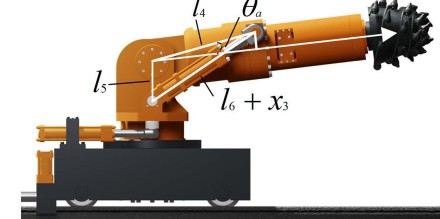

(a) Rotation angle  　　　　　　　　　（b）Swing angle

**Fig 6. Rotation angle and swing angle indicate.**

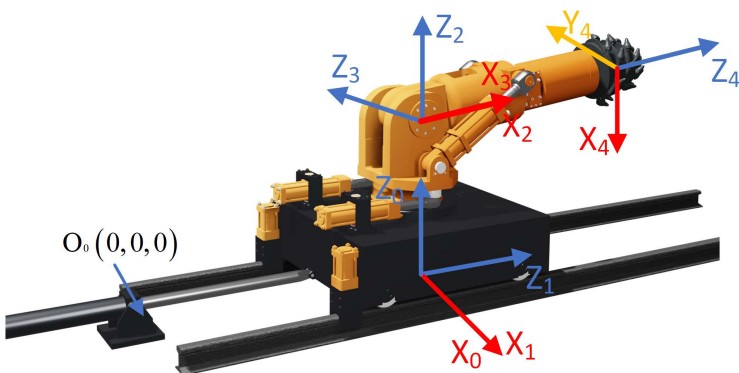

**Fig 7. The coordinate system diagram of the cut part.**

**Table 1. The MDH parameters and joint variables of the cutting part of the boom-type roadheader.**

| Link | $\theta$ | $\alpha$ | $a$ | $d$ |
|---|---|---|---|---|
| 1 | 0 | 90° | 0 | $x_1$ (variate) |
| 2 | $\theta_2$ (variate) | −90° | 0 | $d_2$ |
| 3 | $\theta_3$ (variate) | 90° | 0 | 0 |
| 4 | $\theta_4$ (variate) | 90° | 0 | $d_4$ |

Here, $x_1$ is the displacement of the moving cylinder, $A_1$ is the homogeneous transformation matrix of the base, $T_1$ is the kinematic equation of the base, $x_1^c$ is the displacement data of the moving cylinder foundation, $\Delta x_1$ is the displacement change caused by the passive deformation of the moving cylinder. The kinematic equation of the rotating platform is obtained by translating the displacement of the moving cylinder and the displacement of the rotating cylinder:

$$\begin{cases} \begin{bmatrix} x_1 \\ x_2 \end{bmatrix} \rightarrow T_2 = A_1 A_2 = \begin{bmatrix} c\theta_2 & -s\theta_2 & 0 & 0 \\ s\theta_2 & c\theta_2 & 0 & x_1 \\ 0 & 0 & 1 & d_2 \\ 0 & 0 & 0 & 1 \end{bmatrix} \\ \theta_2 = \theta_2^c + \Delta\theta_2 \end{cases} \tag{9}$$

Here, $d_2$ is the distance between the coordinate of the rotating platform and the coordinate of the base, $A_2$ is the homogeneous transformation matrix of rotating platform, is the kinematic $T_2$ equation of rotating platform, $\theta_2^c$ is the angle of the rotating platform obtained by combining the displacement data of the rotating cylinder foundation with Equation 7, $\Delta\theta_2$ is the corrected displacement data of the rotating cylinder and the corrected angle of the rotating platform obtained by Equation 7. The kinematic equation of the swing arm is obtained through the displacement of the moving cylinder, the displacement of the rotating cylinder and the displacement of the swing arm cylinder:

$$\begin{cases} \begin{bmatrix} x_1 \\ x_2 \\ x_3 \end{bmatrix} \rightarrow T_3 = A_1 A_2 A_3 = \begin{bmatrix} c\theta_2 c\theta_3 & -c\theta_2 s\theta_3 & -s\theta_2 & 0 \\ s\theta_2 c\theta_3 & -s\theta_2 s\theta_3 & c\theta_2 & x_1 \\ -s\theta_3 & -c\theta_3 & 0 & d_2 \\ 0 & 0 & 0 & 1 \end{bmatrix} \\ \theta_3 = \theta_3^c + \Delta\theta_3 \end{cases} \tag{10}$$

Here, $A_3$ is the homogeneous transformation matrix of rotating platform, $T_3$ is the kinematic equation of rotating platform, $\theta_3^c$ is the swing arm angle obtained by combining the displacement data of the swing arm cylinder foundation with Equation 7, $\Delta\theta_3$ is the correction angle of the swing arm obtained by correcting the displacement data of the swing arm cylinder and Equation 7. The kinematic equation of the cutting head is obtained through the displacement of the moving cylinder, the displacement of the rotating cylinder, the displacement of the swing arm cylinder and the rotation angle of the cutting head:

$$
\begin{bmatrix} x_1 \\ x_2 \\ x_3 \\ \theta_4 \end{bmatrix} \rightarrow T_4 = A_1 A_2 A_3 A_4 = \begin{bmatrix} c\theta_2 c\theta_3 c\theta_4 - s\theta_2 s\theta_4 & -c\theta_2 c\theta_3 s\theta_4 - s\theta_2 c\theta_4 & -c\theta_2 s\theta_3 & c\theta_2 s\theta_3 d_4 \\ s\theta_2 c\theta_3 c\theta_4 + c\theta_2 s\theta_4 & -s\theta_2 c\theta_3 s\theta_4 + c\theta_2 c\theta_4 & s\theta_2 s\theta_3 & s\theta_2 s\theta_3 d_4 + x_1 \\ -s\theta_3 c\theta_4 & s\theta_3 s\theta_4 & c\theta_3 & c\theta_3 d_4 + d_2 \\ 0 & 0 & 0 & 1 \end{bmatrix}
\tag{11}
$$

Here, $d_4$ is the distance between the coordinates of the cutting head and the coordinates of the swing arm, $A_4$ is the homogeneous transformation matrix of the cutting head, $T_4$ is the kinematic equation of the cutting head. $\theta_4$ is the rotation angle of the cutting head.

Finally, the kinematic equations of individual structural components are integrated with their respective digital twins, establishing a full-machine kinematic model that simulates the pose dynamics of the boom-type roadheader's cutting module.

## 4. Virtual-real fusion experiment system

### 4.1 System framework

The whole system is composed of the physical experiment system of the cutting part of the boom-type roadheader and the virtual experiment system. As shown in (Fig 8). The physical experiment system includes the mechanical structure, the hydraulic power system and the electrical control system; The virtual experiment system consists of key structural components digital twin, communication system and visual operating system. The virtual-real fusion experiment system is built with a visual operating system powered by UE5. The communication system includes the S7 communication protocol and the mysql communication protocol. The digital coal-rock model and component-level digital twins are embedded via C++ deployment modules. Firstly, the deployment blueprint is encapsulated. Next, grid node coordinates and temporal sequence data are extracted and fed into the pre-programmed grid for rendering, where material coloration is governed by stress and deformation data. Finally, input-output variables are constructed for data transmission within the chain-structured digital twin framework, and spatial coordinates of each component are regulated via kinematic equations to complete the virtual boom-type roadheader cutting module.

The system can realize virtual cutting experiment in complex coal-rock environment by adjusting the distribution number and region of cubic coal-rock units in coal-rock. Environmental simulation can simulate the underground cutting light and dust environment by Lumen global lighting system and Niagara particle system in UE5. At the same time, the construction of the virtual experiment system can map the cutting part experiment system to the digital space, improve the self-sensing ability of the cutting part, and can monitor its various states in real time, providing more data support for the virtual-real fusion experiment and control strategy of the whole machine.

In the process of cutting experiment, both the physical experiment system and the virtual experiment system can drive the swing arm to cut by controlling the pressure or displacement of the swing arm cylinder and the rotating cylinder. The whole system uses the electrical control system as the center connecting the physical device and the virtual experiment system. After the sensor detection system collects a variety of sensor data, it transmits it to the virtual experiment system as the input information through the electrical control system. Meanwhile, the control data generated by the statistical calculation of the virtual experiment system is transmitted to the hydraulic power system through the electrical control system

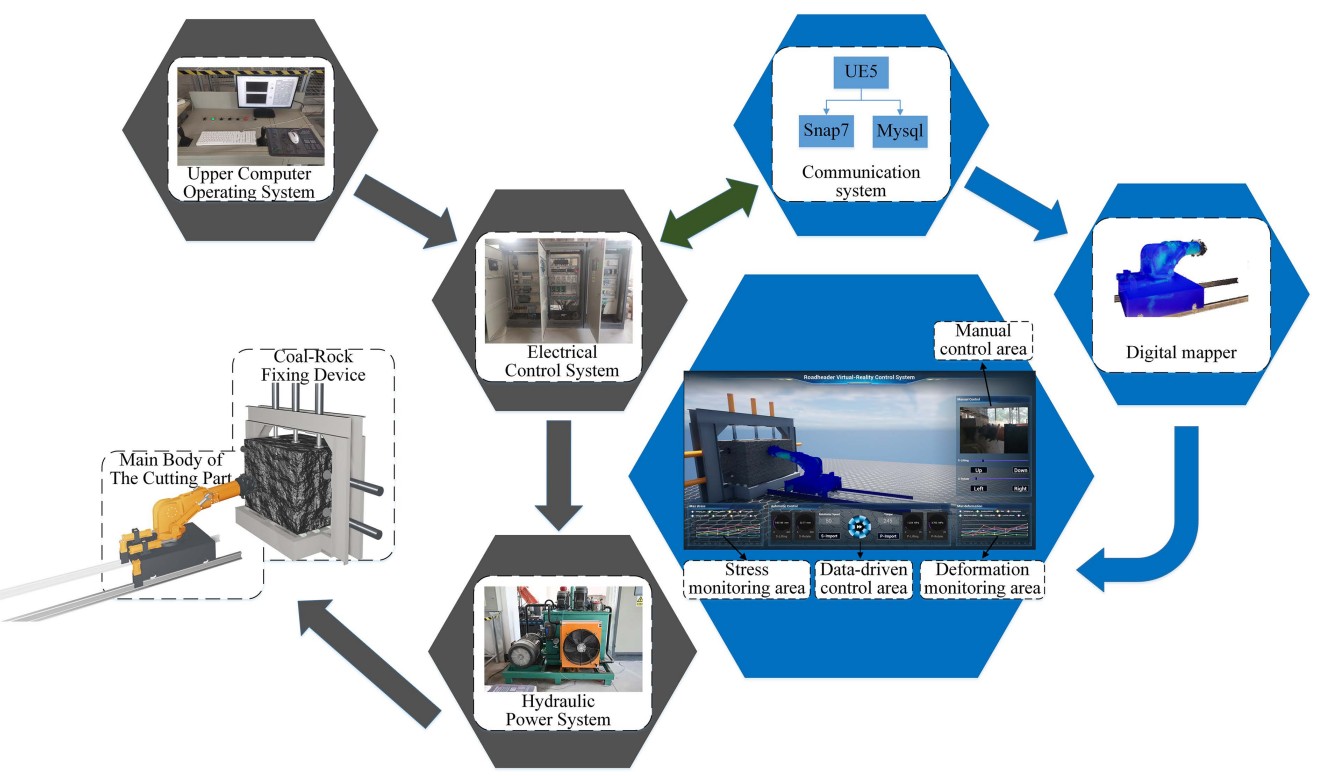

**Fig 8. Virtual-real integration system main body relationship.**

as the control energy flow. The system can realize real-time synchronous linkage between the virtual experiment system and the physical experiment system, modify the virtual data with the physical data, optimize the physical action with the virtual data, and form a virtual-real integrated experiment control system.

## 4.2 System accuracy verification

In order to verify the accuracy of the virtual-real fusion experiment system of the cutting part of the boom-type roadheader, it is necessary to compare the data of the virtual experiment system of the cutting part with the physical experiment system. During the cutting experiment, the load bearing condition of the cutting head in the process of cutting coal-rock is transmitted to the whole body through the swing arm. The swing arm cylinder displacement in the horizontal cutting process and the rotating cylinder displacement in the vertical straight cutting process are the main pose change parameters, and the changes of the two displacement data can reflect the pose and vibration status of the whole machine. Therefore, in this study, virtual cutting of digital coal-rock and physical cutting of solid coal-rock are carried out through the system, and then the displacement change rules of swing arm cylinder and rotating cylinder are compared and analyzed to verify the accuracy of the system. At the same time, the stress data is used to verify the accuracy of the physical performance of the digital mapping system.

The coupling information of coal-rock and cutting tooth is the core of the accuracy of virtual cutting. The selection of hard coal-rock in cutting experiment can enlarge the displacement amplitude of swing cylinder and rotating cylinder, and make the comparison data more convincing. In the process of constructing the digital map of the cutting tooth, the size of coal-rock in the numerical simulation and cutting test is $300 \times 300 \times 300$mm, The size of the cut coal-rock in the

experiment is $2000 \times 1500 \times 1500$mm, Coal-rock cubic unit selected as $100 \times 100 \times 100$mm, The cutting head rotational speed set to$90$r/min. Firstly, the physical experiment system was used to conduct two experiments of horizontal cutting experiment and vertical straight cutting experiment, and the data of swing arm pressure and rotating pressure during the cutting experiment were respectively collected. Then, the data of swing arm pressure and rotating pressure were used as the input pressure of swing arm cylinder and rotating cylinder, and the horizontal cutting and vertical straight cutting experiments were carried out on the virtual experiment system. Considering that the output data of the virtual experiment system is affected by the calculation speed, 100ms is used as the time interval for both the pressure input data and the displacement output data.

The comparative data of the swing arm cylinder displacement change during the horizontal cutting experiment are shown in (Fig 9(a)). The displacement range of the physical swing arm cylinder is 139.0~141.08 mm, Virtual swing arm cylinder displacement range is 138.77~141.10 mm, The average error is 0.188 mm, the physical experiment and the virtual experiment tend to be consistent in the trend of displacement increase and decrease, but there are some errors in amplitude and frequency. (Fig 9(b)). shows the comparative data of displacement change of rotating cylinder during vertical direct cutting experiment. The displacement change range of the physical rotating cylinder is 0.58~2.08 mm. The displacement range of the virtual rotating cylinder is 0.12–1.74 mm, The average error is 0.197 mm, the physical experiment and virtual experiment tend to be consistent in the trend of displacement increase and decrease, and the amplitude and frequency are close to each other. Through the comparative analysis of displacement between horizontal cutting and vertical cutting, the virtual cutting experiment and solid cutting experiment, although there are some errors, can simulate the whole machine position and posture change law and vibration trend during cutting.

The comparative data of stress changes at the bottom of the swing arm cylinder during the horizontal cutting experiment are shown in (Fig 10(a)). The stress changes at the bottom of the physical swing arm cylinder range from 0 to 78.82MPa, and those at the bottom of the virtual swing arm cylinder range from 0 to 89.90MPa, with an average error of 7.63MPa. The comparative data of stress changes at the bottom of the rotating cylinder during the vertical direct cutting experiment are shown in (Fig 10(b)). The stress changes at the bottom of the solid rotating cylinder range from 7 to 47.12MPa, and those at the bottom of the virtual rotating cylinder range from 7.44 to 53.88MPa, with an average error of 4.44MPa.

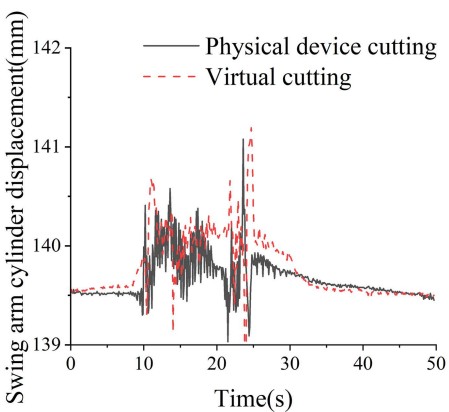
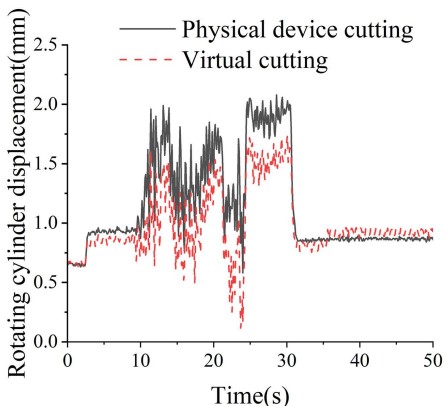

(a) Swing arm cylinder displacement　　　　（b）Rotating cylinder displacement

**Fig 9. Comparison of physical and virtual cutting: (a) Swing arm cylinder displacement;(b) Rotating cylinder displacement.**

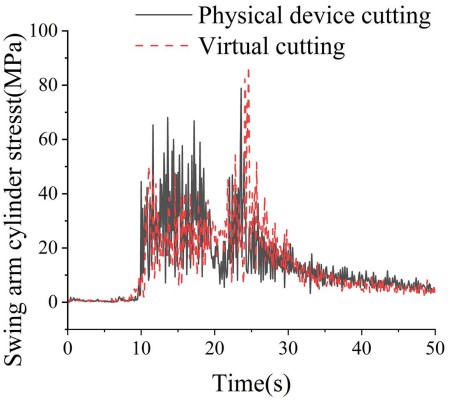
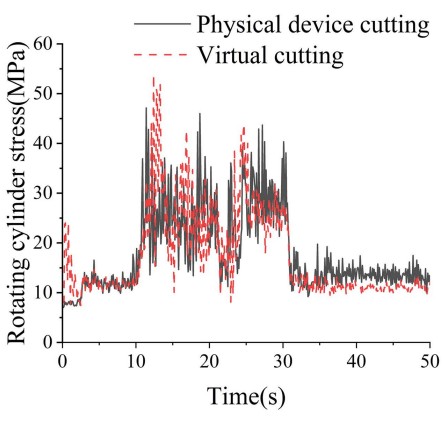

(a) Swing arm cylinder stress （b）Rotating cylinder stress

**Fig 10. Comparison of physical and virtual cutting: (a) Swing arm cylinder stress;(b) Rotating cylinder stress.**

## 5. System feasibility verification

To further validate the feasibility and practical value of the integrated virtual and physical experimental system, this system is utilized as a training platform. The Deep Deterministic Policy Gradient (DDPG) algorithm is employed to optimize the structural stress during the cutting process of the cutting part.

### 5.1. DDPG

DDPG is a combination of deep learning and reinforcement learning algorithm [36]. Centered around the Markov Decision Process (MDP) as its data-driven nucleus, DDPG refines the agent's decision-making procedure through feedback acquisition from the surrounding environment. MDP is described by $s_t$. Here, $s_t$ is the current state of the environment. $s_t$ is the action of the agent in the $s_t$ state. $r_t$ is the reward for the actions of the agent. $s_{t+1}$ is the next transient state of the agent. DDPG primarily comprises Actor network and Critic network. The Critic network assesses the actions produced by the Actor network and subsequently directs the Actor network towards the identification of the optimal control strategy, denoted as $\pi(s_t)$. The goal of the Actor network is to find a control strategy $Q(s_t, a_t)$ to maximize the evaluation value $Q(s_t, a_t)$ of the Critic network by observing the environmental state $\omega\prime$.

The Critic network encompasses both a Critic training network and a Critic target network. The parameters of Critic training network and Critic target network are represented by $\omega\prime$ and $\omega\prime$ respectively. The Critic training network generates the value function $Q_\omega(s_t, a_t)$, which represents the value of the agent's action in the current state. The Critic target network serves as a value function. $Q_{\omega\prime}(s_{t+1}, \pi_{\theta\prime}(s_{t+1}))$., which is dedicated to approximating the value of the agent's action in the subsequent time state. Here, $\pi_{\theta\prime}(s_{t+1})$ represents the action projected for the next time step by the Actor target network:

$$loss = \frac{1}{N}\sum_i (y_i - Q_\omega(s_i, a_i))^2 \tag{12}$$

Here, $N$ is the number of samples. $y_i = r_i + \gamma Q_{\omega\prime}(s_{t+1}, \pi_{\theta\prime}(s_{t+1}))$ is the target value of the action value function in the current state, where $\gamma$ is the coefficient. $a_i = \pi_\theta(s_i) + \varepsilon$ Where $\varepsilon$ is exploration noise. Critic target network adopts soft update mechanism:

$$\omega\prime \leftarrow \xi\omega + (1 - \xi)\omega\prime \tag{13}$$

Here, $\xi$ is the coefficient. The Actor network also contains the Actor training network and the Actor target network. $\varphi$ and $\varphi\prime$ are used to represent the parameters of the Actor training network and the Actor target network, respectively. The Actor training network provides the agent policy $\pi_\varphi(s_t)$ in the current moment state, and the Actor target network provides the policy $\pi_{\theta\prime}(s_{t+1})$ in the next state. The gradient of Actor training network parameter update is:

$$\nabla_{\pi_\varphi} J = \frac{1}{N} \sum_i \nabla_a Q_\omega(s,a)\Big|_{s=s_i, a=\pi_\varphi(s_i)} \nabla_\theta \pi_\varphi(s)\big|_{s_i} \tag{14}$$

Actor target network adopts soft update mechanism:

$$\varphi\prime \leftarrow \xi\varphi + (1-\xi)\varphi\prime \tag{15}$$

## 5.2. Action, environment state and reward function

The control strategy of the cutting part of the boom-type roadheader is to drive the cutting part by controlling the displacement of the swinging arm cylinder and the rotating cylinder. In order to reduce the stress and deformation of the structure in the cutting process, DDPG algorithm is used for intelligent control of swing arm cylinder pressure and rotating cylinder pressure. Therefore, the action output of the agent is set as swing arm cylinder pressure $P_2$ and rotating cylinder pressure $P_3$. In the DDPG algorithm, the state of the environment serves as the foundational basis upon which the agent formulates its decisions. During the cutting process, the following parameters are defined as the environmental state:

1) Speed of cutting drum $n$.

2) Torque of cutting drum $T_R$.

3) Swing arm cylinder displacement $x_2$.

4) Rotating cylinder displacement $x_3$.

The optimization objective of the control strategy is to reduce the structural stress and deformation in the cutting process of the cutting part of the boom-type roadheader while ensuring the completion of the target action, so the reward function is set:

$$r = -\delta(\nabla x_2 + \nabla x_3) - \gamma\sigma_{max} - \eta\varepsilon_{max} \tag{16}$$

Here, $\nabla x_2$ is the difference between the actual displacement of the swing arm cylinder and the target displacement. $\nabla x_3$ is the difference between the actual displacement of the rotating cylinder and the target displacement. $\sigma_{max}$ is the sum of the maximum stress value of each part of the cutting part (If the value is less than 100 MPa, the value is 0). $\varepsilon_{max}$ is the sum of the maximum deformation value of each part of the cutting part (If the value is less than 1 mm, the value is 0). $\delta$, $\gamma$, $\eta$ is the reward coefficient.

## 5.3. Optimization result

The training process is based on DDPG algorithm. 300 times of virtual training and 10 times of virtual-real fusion training are carried out through the cutting part virtual-real fusion experiment system. Virtual training conducts deep reinforcement learning within the virtual experiment system, utilizing virtual sensor data and mechanical parameters (e.g., stress/deformation) to provide environmental states and reward values for the agent, with action strategies executed by the digital twin. Virtual-real fusion training simultaneously operates in both virtual and physical systems. Strain gauges

installed at key positions of the boom-type roadheader's cutting module collect physical stress data, while the physical system acquires sensor data and the virtual system gathers virtual sensor data and mechanical states. Data from both platforms undergo outlier removal and averaging to supply environmental states and reward values, with agent-derived action strategies executed concurrently by the physical entity and its digital twin. The cutting task is linear cutting, and the environment state and reward information in the virtual training process are derived from the calculation data of the virtual experiment system. In the process of virtual-real fusion training, the environment state information comes from the sensor data, and the reward information comes from the calculation data of the virtual experiment system. As shown in (Fig 11).

Both the Actor Network and the Critic network use a two-layer Artificial Neural Network (ANN), each containing 64 units. In order to verify the optimization results of the control strategy, the same coal-rock is cut 5 times by the control strategy before and after optimization respectively, and the data of the swing arm cylinder strain gauge is collected and compared with the previous cut without the control strategy. (Fig 12(a)) shows the normalized cumulative reward in the training process, and (Fig 12(b)) shows the optimization comparison results. Prior to the optimization of the control strategy, the stress values recorded by strain gauges during coal rock cutting operations exceeded 100MPa for 58.80% to 68.25% of the total operational time. following adjustments and enhancements to the control strategy, there was a notable reduction in the incidence of strain gauge stress values surpassing the limit of 100MPa during coal rock cutting operations, with the proportion falling to a range of 45.15% to 54.60%. for coal rock samples with similar physical characteristics (identified under the same sequence number), the optimized control strategy effectively decreased the frequency of stress exceeding the 100MPa threshold by approximately 10%.

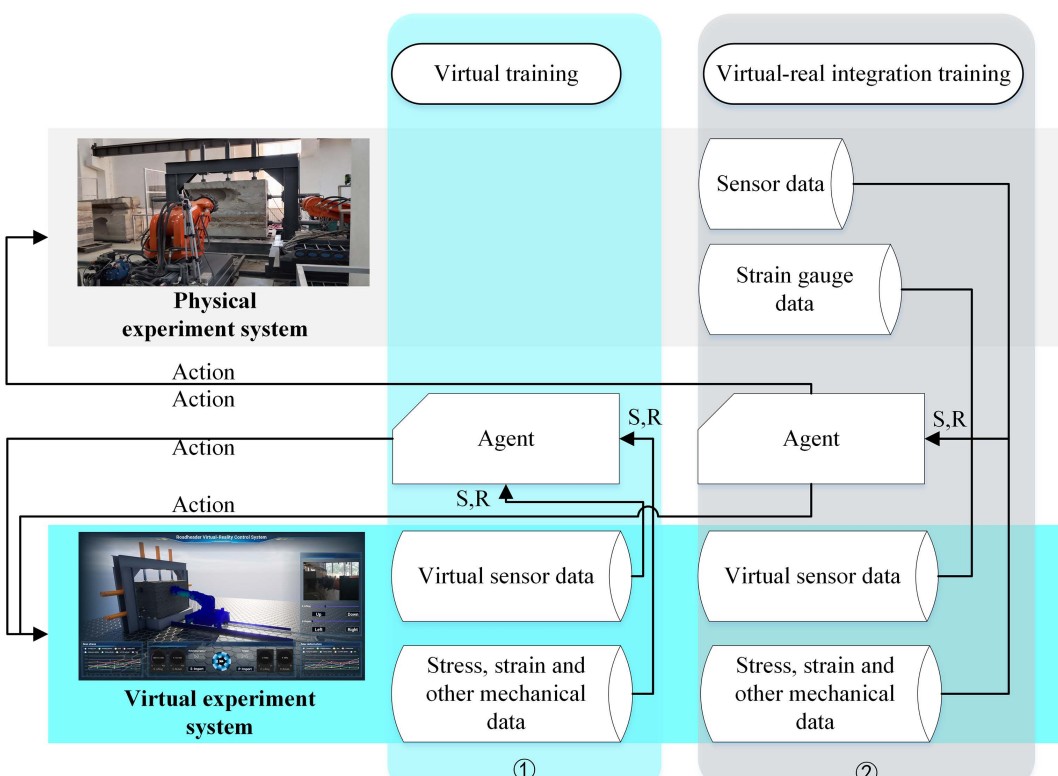

**Fig 11. Physical experiment system and virtual experiment system.**

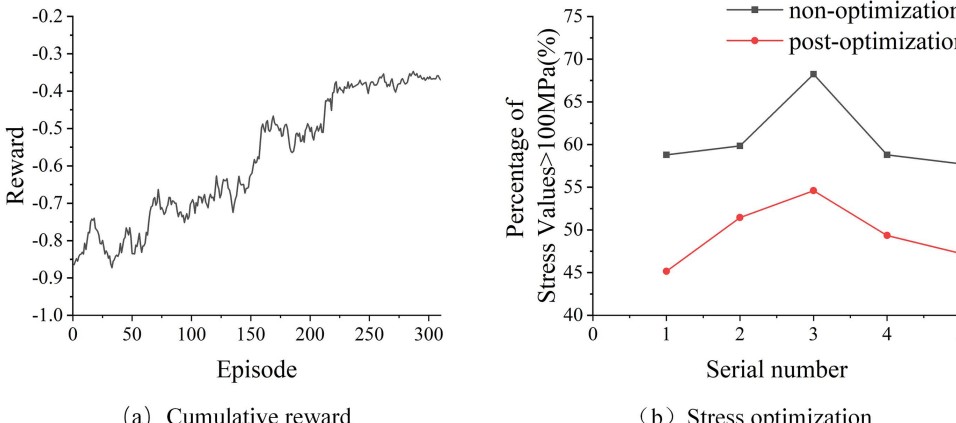

(a) Cumulative reward (b) Stress optimization

**Fig. 12. DDPG optimization results: (a) Cumulative reward; (b) Stress optimization.**

The results show that adopting the DDPG algorithm for extensive virtual training supplemented by targeted physical validation within the virtual-real fusion system successfully optimized the cutting module's structural stress control in the cutting module (Fig 13). This process validated the system's capability to construct homogeneous/heterogeneous coal-rock digital models through hard coal-rock mechanical simulation. During hard coal-rock cutting simulations, average stress errors below 7.63 MPa in swing arm cylinders and rotary cylinders confirmed accurate identification and fitting of cutting tooth-coal-rock coupling relationships. Real-time monitoring of digital twins and physical entities was achieved through synchronized physical/virtual sensor data acquisition. High-fidelity pose simulation and mechanical data output demonstrated effective real-time motion/environment simulation with whole-machine multi-modal perception. The 10% structural stress reduction validated cross-domain virtual-real data fusion for enhanced experimental dataset accuracy.

If the above process is carried out through traditional physical experiments, there will be risks of accidents caused by the actions being tested. In terms of resource consumption, it would roughly take 2–3 years to cut 52 pieces of coal-rock with dimensions of 2000×1500×1500 mm. The entire process of completing the above-mentioned tasks using the virtual-real fusion experimental system lasted for a total of 5 months, including a 3-month preparation period and a 2-month training period. During the preparation period, numerical simulation was the main method, and physical experiments only used 10 pieces of 300×300×300 mm coal-rock. During the training period, virtual training could conduct any number of cutting tests, and the virtual-real integration test used 2 pieces of 2000×1500×1500 mm coal-rock. In conclusion, the virtual-real fusion experimental system can reduce the safety risks in cutting tests, enhance the efficiency of the tests, and lower the testing costs.

## 6. Conclusions

This study has successfully developed a virtual-real fusion experimental system for boom-type roadheader cutting modules, demonstrating three core capabilities:

(1) Aiming at the problems of high experimental research cost and safety risk of boom-type roadheaders, a virtual-real fusion experiment system based on chain-type digital mapping bodies is proposed. This system is applicable to virtual experiments, physical entity experiments and virtual-real fusion experiments of the cutting part of the boom-type roadheaders. This system takes Virtual-Dominant Entity Calibration as its functional core and is capable of constructing digital models of coal-rock, simulating mechanical properties of structures, and conducting virtual-real fusion experiments. Subsequently, the DDPG algorithm was adopted to conduct optimization experiments on the control strategy of the cutting part, verifying the feasibility of the system.

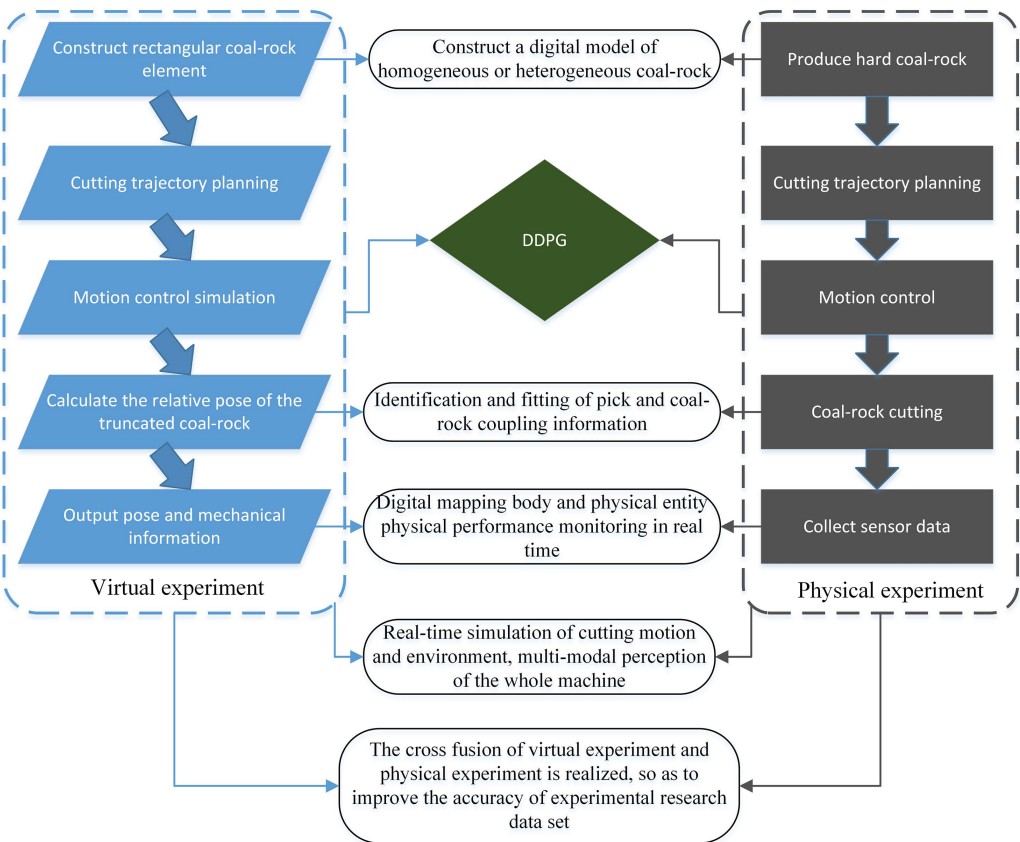

**Fig 13. Functional proof of the virtual-real fusion experiment system.**

(2) Based on deep learning, coal-rock cutting experiments, and numerical simulation, a discretized cubic unitization recognition method for coal-rock is proposed, and a digital twin of coal-rock for the load of the cutting teeth is constructed. This method can construct digital coal-rock, provide mechanical load input for virtual cutting experiments of the roadheader, realize digital simulation of the complex coal-rock cutting process by the cutting head, and reduce the difficulty and cost of fabricating large pieces of coal-rock in cutting experiments.

(3) Based on coordinate space transformation and ROM technology, a chained digital twin model is proposed. Through data transmission and coordinate transmissions, the pose and mechanical state changes of the cutting part of the boom-type roadheader can be simulated. The overall digital mapping of complex assemblies has been achieved, providing a basic platform for virtual experiments.

The next step is to improve the coal-rock digital mass by studying the mechanical properties, degradation mechanisms, and fracture mechanisms of the surrounding rock to increase accuracy, applying this system to other tunnel boring machines or mining equipment to form a systematic framework, to solve problems of large-scale mechanical testing, and to combine advanced technologies such as artificial intelligence and big data to further enhance the intelligence level of the system.

## Supporting information

**S1 Table. Data of physical cutting and virtual cutting.**
(XLSX)

**S2 Table. Reward curve data and stress optimization data.**
(XLSX)

**S3 Python code. DDPG.**
(RAR)

**S4 Video. Simulation effect of the cutting part.**
(MP4)

## Author contributions

**Conceptualization:** Xiaoyu Ding.

**Data curation:** Chuanxu Wan.

**Formal analysis:** ce chen.

**Funding acquisition:** Jianzhuo Zhang.

**Investigation:** Jianzhuo Zhang, ce chen.

**Methodology:** Tao Wang.

**Resources:** Jianzhuo Zhang, Tao Wang, Wenliang Li.

**Supervision:** Xiaoyu Ding, Chuanxu Wan.

**Validation:** Tao Wang, Wenliang Li.

**Writing – original draft:** Jianzhuo Zhang, ce chen.

**Writing – review & editing:** Xiaoyu Ding.

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
