## [Decision Letter · Decision Letter 0]

16 May 2025

Dear Dr. chen,

Thank you for submitting your manuscript to PLOS ONE. After careful consideration, we feel that it has merit but does not fully meet PLOS ONE’s publication criteria as it currently stands. Therefore, we invite you to submit a revised version of the manuscript that addresses the points raised during the review process.

We look forward to receiving your revised manuscript.

Kind regards,

Zhengzheng Cao

Academic Editor

PLOS ONE

 [The project is supported by National Key Research and Development Program of China - Intelligent impact support technology and equipment foiimpact hazardous tunnels and engineering demonstration(2022YFC3004605)].

4. In the online submission form, you indicated that [The data that support the findings of this study are available on request from the corresponding author, upon reasonable request.].

Additional Editor Comments:

**Comments from PLOS Editorial Office** : We note that one or more reviewers has recommended that you cite specific previously published works. As always, we recommend that you evaluate the requested works to determine whether they are relevant and should be cited. It is not a requirement to cite these works and you may remove any added citations before the manuscript proceeds to publication. We appreciate your attention to this request.

Reviewers' comments:

Reviewer's Responses to Questions

**Comments to the Author**

1. Is the manuscript technically sound, and do the data support the conclusions?

Reviewer #1: Partly

Reviewer #2: Yes

Reviewer #3: Yes

2. Has the statistical analysis been performed appropriately and rigorously?

Reviewer #1: Yes

Reviewer #2: Yes

Reviewer #3: Yes

3. Have the authors made all data underlying the findings in their manuscript fully available?

Reviewer #1: Yes

Reviewer #2: Yes

Reviewer #3: Yes

4. Is the manuscript presented in an intelligible fashion and written in standard English?

Reviewer #1: Yes

Reviewer #2: Yes

Reviewer #3: Yes

Reviewer #1: This paper focuses on the problems such as the high risk of underground tests for boom roadheaders and the high cost of ground tests. It innovatively proposes a virtual-real fusion test system for the cutting part of the roadheader, constructs a digital mapping of the cutting part of the roadheader, and realizes the organic integration of the physical equipment test and the virtual test. It has certain innovation and high engineering application value. However, there are still the following issues in the paper that need to be revised and improved.

1�In the process of constructing the rectangular coal-rock unit in Section 2.1 of the paper to form the input matrix of the digital mapping of the pick load, the author considered parameters such as the depth, angle, and speed of the pick cutting into the coal-rock, as well as the damage degree of the cut coal-rock unit. However, the author did not elaborate in detail on the reasons for selecting these parameters. It is recommended that the author supplement relevant content and explain in detail the necessity and rationality of selecting these parameters.

2�In Fig.2 of Section 2.1 of the paper, the coordinate colors are dim, which may affect the readers' understanding of the chart content. It is recommended that the author adjust the coordinate colors to make them more eye-catching and clearly mark the main parameters to enhance the readability of the chart.

3�In Section 3.2 of the paper, the Denavit-Hartenberg notation (DH method) is used for the motion equations, but the DH parameters are not clearly given, and there is a lack of a detailed description of the specific way the pose matrix is used in the test system. It is recommended to add relevant explanations so that readers can better understand the relevant content.

4�For the acquisition of the pose information and mechanical information of the virtual-real fusion test system, it is necessary to clearly state what types of sensors are used, the specific measurement methods, and the arrangement of the sensors. At the same time, it is also necessary to explain how to use the local and limited sensor data to reflect the mechanical characteristics of the whole machine in different poses.

5�Section 5.3 of the paper describes the sources of the environmental state and reward information during the virtual training process and the virtual-real fusion training process. During the virtual training, both the environmental state and reward information come from the calculation data of the virtual test system. In the virtual-real fusion training process, the environmental state information comes from the sensor data, and the reward information comes from the calculation data of the virtual test system. In order to display these data information more clearly, it is recommended that the author present them intuitively in the form of a chart for easy comparative analysis.

6�The number of references cited is relatively small. It is recommended that the author appropriately increase the number of references cited, widely draw on the research achievements in related fields, and further enrich the content of the article.

Reviewer #2: (1)In the introduction section of the article, only some shortcomings of traditional research methods are listed, with limited comparative analysis of recent achievements by other scholars in the improvement of tunnel boring machine experimental systems or similar applications of virtual-real fusion technology. It is recommended to supplement relevant content to more clearly highlight the unique innovations of this study.

(2)The stress values in the text use incorrect units, such as 100mpa.

(3)After optimization, the stress overrun ratio decreases by approximately 10%. To demonstrate the superiority of DDPG, comparisons with other algorithms can be conducted.

(4)Some latest research work related with this TOPIC can be referred. Study on the degradation mechanism of mechanical properties of red sandstone under static and dynamic loading after different high temperatures. 

(5)The entire text contains numerous language errors, such as "100sm" and "DDGP". Please carefully check the entire text and make necessary revisions.

(6)In the conclusion, avoid restating methodological details and instead focus on highlighting innovative aspects and future research directions.

Reviewer #3: 1. This paper proposes a virtual-reality fusion experimental system for the cutting part of a cantilevered roadheader, which has important innovations in reducing the difficulty of the experiment and improving the adaptability of the cutting. However, it is recommended to further clarify the actual application scenarios and potential socio-economic value of the system to highlight the innovation and practical contribution of its research.

2. This paper introduces the construction process of the system in detail, including key technologies such as coal-rock discretization, digital mapping body construction, and virtual-reality system fusion. However, it is recommended to add more technical implementation details, such as the implementation steps, parameter settings, and code implementation of specific algorithms, so that readers can better understand and reproduce the research content.

3. This paper verifies the accuracy of the virtual-reality fusion experimental system through experiments and demonstrates its effectiveness in reducing the stress of the cutting part. However, it is recommended to further analyze the stability and reliability of the experimental data, such as conducting multiple repeated experiments and using different coal-rock samples for experiments, so as to enhance the persuasiveness of the experimental conclusions.

4. The virtual-reality fusion experimental system performs well in real-time and synchronization, but it is recommended to further explore the scalability and maintainability of the system. For example, how to quickly adapt to different types of tunnel boring machines, how to conveniently upgrade and maintain the system, etc. In addition, it is possible to consider introducing more optimization algorithms to improve the overall performance of the system.

5. The abstract of the article needs to be revised, and the importance of the article should be highlighted. There are insufficient references, so more references need to be supplemented. There are too few references, which need to be supplemented to 30. The background and mechanism are not introduced clearly.

Mechanical behavior and fracture mechanism of high-temperature granite cooled with liquid nitrogen for geothermal reservoir applications. Physics of Fluids 2025; 37 (2): 026616. https://doi.org/10.1063/5.0253668

6. The virtual-real fusion experimental system proposed in the article has broad application prospects, especially in the fields of coal mining and machinery manufacturing. It is recommended that the author further explore the application potential and promotion strategy of the system, and clarify the future research direction. For example, we can study how to apply the system to other types of tunnel boring machines or mining equipment, or explore how to combine advanced technologies such as artificial intelligence and big data to further improve the intelligence level of the system.

**Do you want your identity to be public for this peer review?** For information about this choice, including consent withdrawal, please see our Privacy Policy

Reviewer #1: No

Reviewer #2: No

Reviewer #3: No

---

## [Author Response · Author response to Decision Letter 1]

7 Jun 2025

Thank you to the editor and the review experts for the review of the manuscript. Regarding the questions and opinions raised, we have given detailed responses. However, since only text can be uploaded here in the system and formulas, pictures and tables cannot be uploaded, we have uploaded the response letter file. The responses to the relevant questions are in the response letter file. Please excuse me.

---

## [Decision Letter · Decision Letter 1]

16 Jul 2025

Dear Dr. chen,

Thank you for submitting your manuscript to PLOS ONE. After careful consideration, we feel that it has merit but does not fully meet PLOS ONE’s publication criteria as it currently stands. Therefore, we invite you to submit a revised version of the manuscript that addresses the points raised during the review process.

**The authors must outline the novelty points of this paper with proposed model. Additionally, the discussion should be enlarged to analyze and compare with previous research results. **

We look forward to receiving your revised manuscript.

Kind regards,

Tien Anh Tran

Academic Editor

PLOS ONE

Journal Requirements:

Reviewers' comments:

Reviewer's Responses to Questions

**Comments to the Author**

Reviewer #2: All comments have been addressed

Reviewer #3: (No Response)

Reviewer #4: All comments have been addressed

2. Is the manuscript technically sound, and do the data support the conclusions?

Reviewer #2: Yes

Reviewer #3: (No Response)

Reviewer #4: Yes

3. Has the statistical analysis been performed appropriately and rigorously?

Reviewer #2: Yes

Reviewer #3: (No Response)

Reviewer #4: Yes

4. Have the authors made all data underlying the findings in their manuscript fully available?

Reviewer #2: Yes

Reviewer #3: (No Response)

Reviewer #4: No

5. Is the manuscript presented in an intelligible fashion and written in standard English?

Reviewer #2: Yes

Reviewer #3: (No Response)

Reviewer #4: Yes

Reviewer #2: it can be accepted.

Aiming at the problems of high safety risks, economic costs, and inefficiency in experimental research on boom-type roadheaders, this study proposes a virtual-real fusion system for the cutting module. This system incorporates functions including digital modeling of coal-rock, simulation of echanical properties of the cutting unit, and integration of virtual and physical experiments.

Reviewer #3: (No Response)

Reviewer #4: This study presents an innovative virtual-real fusion system for roadheader cutting experiments, integrating digital twins, DDPG optimization, and coal-rock discretization. While the concept is promising and initial validation shows moderate accuracy, critical methodological gaps persist in coal-rock damage modeling, system generalizability, and cost-benefit justification. Addressing these would strengthen industrial applicability and scientific rigor.

1. The introduction focuses primarily on coal mine applications. To strengthen the rationale, explicitly reference non-coal applications of boom-type roadheaders (e.g., civil tunnel construction, hydropower caverns, or urban underground infrastructure (https://doi.org/10.1016/j.scs.2022.104369;
https://doi.org/10.1016/j.tust.2023.105382)) and cite recent studies demonstrating their use in diverse geotechnical settings (e.g., hard rock tunneling in metro projects). This would better justify the system’s universal value beyond coal-centric scenarios.

2. Expand testing beyond hard coal-rock at fixed RPM (90 rpm). Include diverse coal-rock strengths (soft/heterogeneous strata) and dynamic operational scenarios (variable cutting speeds, intermittent loads) to demonstrate system robustness. Quantify the impact of 100ms latency on real-time control stability.

3. Clarify the deep learning architecture (e.g., layer types, node counts, optimization algorithms) and validation metrics (e.g., RMSE, R²) used to derive the damage coefficient (Eq. 4). Justify why a 3×3×3 boundary matrix (Eq. 5) suffices for spatial heterogeneity modeling.

4. Explicitly compare resource requirements (time, materials, labor) between traditional physical experiments and the proposed virtual-dominant approach (e.g., 300 virtual + 10 physical DDPG runs). Estimate savings in coal-rock fabrication, sensor deployment, and downtime to highlight scalability.

**Do you want your identity to be public for this peer review?** For information about this choice, including consent withdrawal, please see our Privacy Policy

Reviewer #2: No

Reviewer #3: No

Reviewer #4: No

---

## [Author Response · Author response to Decision Letter 2]

24 Jul 2025

Thank you to the editor and the review experts for the review of the manuscript. Regarding the questions and opinions raised, we have given detailed responses.However, since only text can be uploaded here in the system and formulas cannot be uploaded, we have uploaded the response letter file. The responses to the relevant questions are in the response letter file. Please excuse me.

---

## [Decision Letter · Decision Letter 2]

30 Jul 2025

Research on a virtual-real fusion experimental system for the cutting part of a boom-type roadheader

PONE-D-25-15395R2

Dear Dr. chen,

We’re pleased to inform you that your manuscript has been judged scientifically suitable for publication and will be formally accepted for publication once it meets all outstanding technical requirements.

Kind regards,

Tien Anh Tran

Academic Editor

PLOS ONE

Additional Editor Comments (optional):

Reviewers' comments:

Reviewer's Responses to Questions

**Comments to the Author**

Reviewer #2: All comments have been addressed

Reviewer #3: All comments have been addressed

Reviewer #4: All comments have been addressed

2. Is the manuscript technically sound, and do the data support the conclusions?

Reviewer #2: Yes

Reviewer #3: (No Response)

Reviewer #4: Yes

3. Has the statistical analysis been performed appropriately and rigorously?

Reviewer #2: Yes

Reviewer #3: (No Response)

Reviewer #4: Yes

4. Have the authors made all data underlying the findings in their manuscript fully available?

Reviewer #2: Yes

Reviewer #3: (No Response)

Reviewer #4: Yes

5. Is the manuscript presented in an intelligible fashion and written in standard English?

Reviewer #2: Yes

Reviewer #3: (No Response)

Reviewer #4: Yes

Reviewer #2: Aiming at the problems of high safety risks, economic costs, and inefficiency in experimental research on boom-type roadheaders, this study proposes a virtual-real fusion experimental system for the cutting module. This system incorporates functions including digital modeling of coal-rock, simulation of mechanical properties of the cutting unit, and integration of virtual and physical experiments.

ACCEPT

Reviewer #3: accept the paper Please use the space provided to explain your answers to the questions above. You may also include additional comments for the author, including concerns about dual publication, research ethics, or publication ethics. (Please upload your review as an attachment if it exceeds 20,000 characters) (Limit 100 to 20000 Characters)

Reviewer #4: The authors have thoroughly addressed reviewer feedback (e.g., clarifying deep learning architectures, quantifying resource savings vs. physical tests, and broadening non-coal applications). Revisions demonstrate robustness, with latency analysis (<100 ms) confirming real-time stability. The work significantly advances intelligent excavation research by enabling virtual-dominant, cost-effective experimentation. All critical revisions are complete, and the manuscript now meets publication standards. Recommend acceptance.

**Do you want your identity to be public for this peer review?** For information about this choice, including consent withdrawal, please see our Privacy Policy

Reviewer #2: No

Reviewer #3: No

Reviewer #4: No

---

## [Editor Report · Acceptance letter]

PONE-D-25-15395R2

PLOS ONE

Dear Dr. chen,

I'm pleased to inform you that your manuscript has been deemed suitable for publication in PLOS ONE. Congratulations! Your manuscript is now being handed over to our production team.

Kind regards,

on behalf of

Professor Tien Anh Tran

Academic Editor

PLOS ONE